# Development, manufacturing, and preliminary validation of a reusable half-face respirator during the COVID-19 pandemic

Vahid Anwari[1,2], William C. K. Ng[3,4,5] *, Arnaud Romeo Mbadjeu Hondjeu[4], Zixuan Xiao[6‡], Edem Afenu[7,8‡], Jessica Trac[8‡], Kate Kazlovich[2,9‡], Joshua Hiansen[2‡], Azad Mashari[2,3,4]

1 Joint Department of Medical Imaging, Toronto General Hospital, University Health Network, Toronto, Ontario, Canada, 2 The Lynn and Arnold Irwin Advanced Perioperative Imaging Lab, Department of Anesthesiology and Pain Management, Toronto General Hospital, University Health Network, Toronto, Ontario, Canada, 3 Department of Anesthesiology and Pain Medicine, University of Toronto, Toronto, Ontario, Canada, 4 Department of Anesthesiology and Pain Management, Toronto General Hospital, University Health Network, Toronto, Ontario, Canada, 5 Department of Anaesthesia and Pain Management, The Hospital for Sick Children, Toronto, Ontario, Canada, 6 Faculty of Medicine, University of Alberta, Edmonton, Alberta, Canada, 7 School of Biomedical Engineering, University of Toronto, Toronto, Ontario, Canada, 8 Faculty of Medicine, University of Toronto, Toronto, Ontario, Canada, 9 Institute of Biomaterials and Biomedical Engineering, University of Toronto, Ontario, Canada

☯ These authors contributed equally to this work.
‡ These authors also contributed equally to this work.
* DrWilliam.Ng@mail.utoronto.ca

**Data Availability Statement:** Data will be available on repository https://github.com/tgh-apil/Reusable-N95-Respirator/blob/master/10-

## Abstract

### Introduction

The COVID-19 pandemic has led to widespread shortages of N95 respirators and other personal protective equipment (PPE). An effective, reusable, locally-manufactured respirator can mitigate this problem. We describe the development, manufacture, and preliminary testing of an open-hardware-licensed device, the "simple silicone mask" (SSM).

### Methods

A multidisciplinary team developed a reusable silicone half facepiece respirator over 9 prototype iterations. The manufacturing process consisted of 3D printing and silicone casting. Prototypes were assessed for comfort and breathability. Filtration was assessed by user seal checks and quantitative fit-testing according to CSA Z94.4–18.

### Results

The respirator originally included a cartridge for holding filter material; this was modified to connect to standard heat-moisture exchange (HME) filters (N95 or greater) after the cartridge showed poor filtration performance due to flow acceleration around the filter edges, which was exacerbated by high filter resistance. All 8 HME-based iterations provided an adequate seal by user seal checks and achieved a pass rate of 87.5% (N = 8) on quantitative testing, with all failures occurring in the first iteration. The overall median fit-factor was

Publication/PLOS_manufacturing_deidentified_12.15.2020.xlsx.

**Funding:** The authors received no specific funding for this work.

**Competing interests:** The authors have declared that no competing interests exist.

1662 (100 = pass). Estimated unit cost for a production run of 1000 using distributed manufacturing techniques is CAD $15 in materials and 20 minutes of labor.

## Conclusion

Small-scale manufacturing of an effective, reusable N95 respirator during a pandemic is feasible and cost-effective. Required quantities of reusables are more predictable and less vulnerable to supply chain disruption than disposables. With further evaluation, such devices may be an alternative to disposable respirators during public health emergencies. The respirator described above is an investigational device and requires further evaluation and regulatory requirements before clinical deployment. The authors and affiliates do not endorse the use of this device at present.

## Introduction

The COVID-19 pandemic has unveiled widespread shortages of N95 respirators worldwide. In fact, all threats of pandemic disease in the last 20 years have resulted in either local or global shortages of disposable N95s [1, 2]. This latest pandemic has once again highlighted the inadequacy of existing strategies for pandemic respiratory protection planning [3]. A 2015 modeling study done jointly by the National Center for Immunization and Respiratory Diseases and the Office of Infectious Diseases, Centers for Disease Control and Prevention (CDC) in the United States [4] estimated that a respiratory pandemic affecting 20–30% of the US population would require between 1.7 and 7.3 billion disposable respirator masks, more than *30 times* the total national holdings in the US at the time. Other studies have reached similar conclusions and their predictions have been largely borne out by the current pandemic [5, 6].

Health care workers (HCWs) face hazardous occupational exposures to infectious organisms, many of which are spread through airborne or aerosol routes [5]. To minimize risk of infection when treating patients with COVID-19, the CDC recommends the use of personal protective equipment (PPE) including gown, gloves, N95 respirator, and a face-shield or goggles. Paramount to emergency preparation is accessibility of PPE, particularly respiratory protective devices (RPDs). RPDs and other PPE are the last line of defense when exposures cannot be reduced to an acceptable level using other control methods [7]. Disposable N95 filtering face-piece respirators are currently the most commonly used devices to protect HCWs [8]. These respirators are designed to create a seal on the face and, when sealed, remove at least 95% of airborne particles of size around 0.3 um (the most penetrating particle size) [9, 10].

In the long run, addressing the failures of respiratory protection strategies will require a holistic, systems-based approach of which technical innovations in respirator design and production will be only a part. In the midst of the crisis however, technical innovation is often the most accessible point of intervention for addressing acute local needs. We describe the development, manufacturing process and initial performance evaluation of a reusable N95 respirator, referred to as SSM ("Simple Silicone Mask") hereafter, that can be manufactured on-site, using distributed manufacturing technologies. Such reusable, locally manufacturable devices are far less vulnerable to sudden surges in demand and supply chain disruptions that accompany global pandemics. The device took inspiration from open-source designs and was developed through the collaboration of physicians, engineers, researchers, students, and private sector partners. It is licensed under a CERN Open Hardware Version 2-Strongly Reciprocal

license, which permits modification and production of the device without specific permission, including for industrial manufacturing, commercial distribution and sale.

## Methods

### Model 1: Rigid 3D-printable mask with filter cartridge

Our design process began with the "Stop-Gap Face Mask (SFM)" (Fig 1) an open-source licensed design by Chris Richburg available through the National Institutes of Health's 3D Print Exchange (https://3dprint.nih.gov/discover/3dpx-013429). This mask consists of a rigid 3D-printable body with a built-in cartridge frame designs to hold flat sheets of filter material. High-resolution prototypes were created using PolyJet printing on the Stratasys® J750 Digital Anatomy™ using Digital ABS Plus resin. Based on a review of scientific and lay literature, discussions with material science experts and industry partners, as well as product availability during the current pandemic, the following materials were selected for testing: 5N11 (3M, Maplewood, MN, USA) and H400 (Halyard Health, Alpharetta, GA, USA).

A sample of each filter material measuring 5 cm$^2$ (effective surface area ~ 16 cm$^2$; 0.5–1.0 mm thick) was fitted onto the SFM filter cartridge holder. The respirator was attached to a test fixture and sealed with aluminum tape (Figs 2 and 3). Samples were tested at a flow of 30 +/- 4 L/min using 2% sodium chloride solution in distilled water (NaCl), according to US Government 42 Code of Federal Regulations (CFR) Part 84 Test for RPD [11, 12]. This regulation on respiratory protection equipment is informed by the research and guidance by the National Institute for Occupational Safety and Health (NIOSH). The test sequentially measures the filtration of 10–400 nm NaCl particles. General purpose health care respirators must achieve 95% filtration of particles around 0.3 um [10]. Particle count (count / cm$^3$) and resistance (kPa) around mask and filter seal were measured.

The filter-cartridge design showed very poor filtration performance (see Table 1), regardless of the filter materials used. Failure was largely due to the inability to create a reliable seal around the removable filter within the filter-cartridge, despite several revisions of the design.

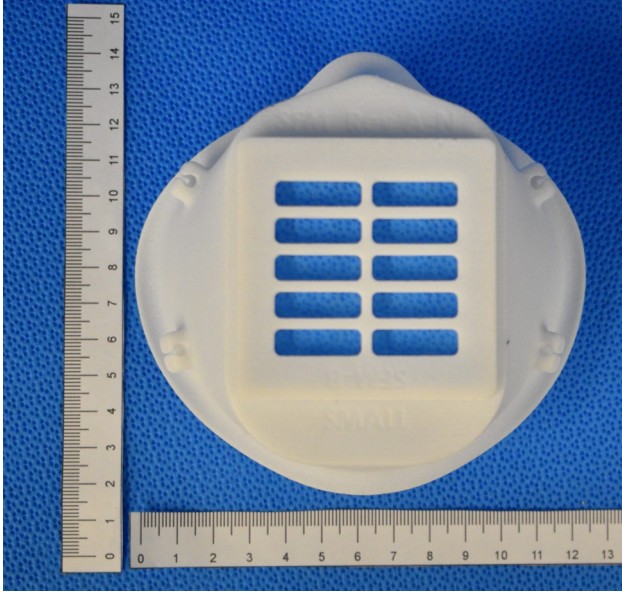

**Fig 1. Initial respirator design "SFM" with a sample of the filter material in the single layer cartridge configuration.**

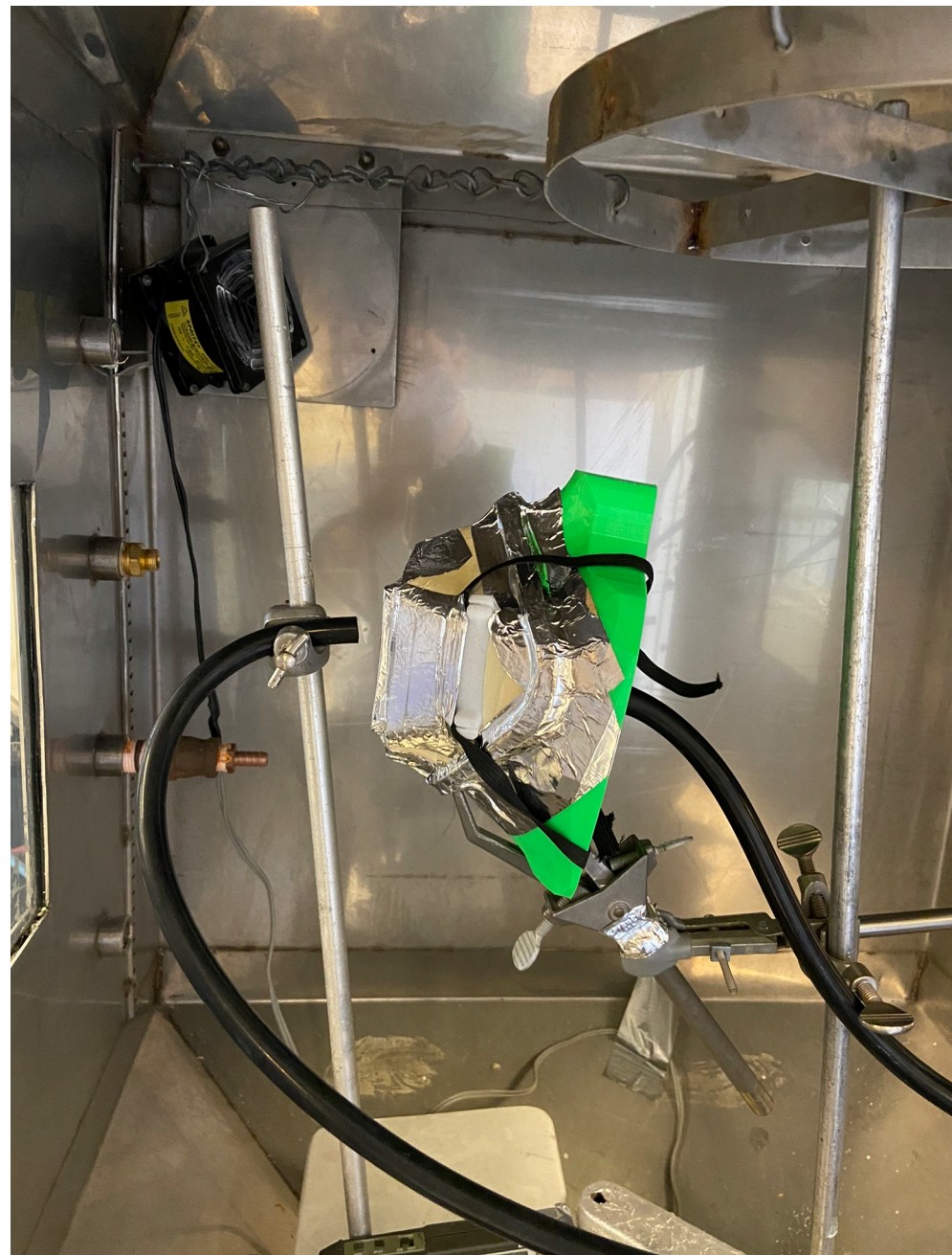

**Fig 2. In-profile representation of the SFM respirator assembled on a fixture housed in the exposure chamber for filtration efficiency testing.**

In addition, the 3D printed body was unlikely to be comfortable for long wear and required significant print time for each unit.

The filter-cartridge style design was thus abandoned, in favor of masks connected directly to off-the-shelf Heat and Moisture Exchange (HME) filters with known filtration efficiency and pressure drop properties. The team then moved on to a model 2, using silicone for the mask body that incorporates an HME filter.

**Fig 3. Schematic of preliminary filtration efficiency testing [13].** FFR stands for face filtering respirator. EC 3080 is an electrostatic classifier and CPC 3785 a condensation particle counter.

## Model 2: Silicone mask cast from 3D-printed mold connected to off-the-shelf HME filter

The starting point for the final SSM design was the Simple Silicone Respirator (SSR) designed by Dr. Christian Petropolis at the University of Manitoba. The SSR was cast from 3D printed molds, which allowed use of a wider range of materials than direct 3D printing and also permitted more rapid scale-up of production for the final design.

At each design iteration, the device and corresponding mold designs were modified using computer-aided design (CAD) software (Onshape, www.onshape.com; and Fusion 360™, Autodesk®). The final 4-part mold design is shown in Fig 4.

Each part was exported from the CAD software as a stereolithography (STL) file and prepared for 3D printing using the PrusaSlicer (v. 2.2.0) software. Molds were printed on Prusa I3 MK3S (Prusa Research, Prague, Czech Republic) 3D printers with a 0.4 mm nozzle using PLA (polylactic acid) or PETG (polyethylene terephthalate glycol); Figs 5–7 depicts the mold parts.

To secure the mask to the face, a harness was designed and printed. Figs 8 and 9 represents the harness. The STL files for the molds and harness are available at our project repository at https://github.com/tgh-apil/Reusable-N95-Respirator.

To make the respirator body, the 3D printed molds from Figs 5–7 were assembled for silicone pouring (Fig 10). A 2-part biocompatible casting silicone (Dragon Skin™20, Smooth-on, Macungie, PA, USA) [14] was poured into the assembled mold from the pour hole shown in Fig 7. The mold was then allowed to cure for over 4 hours. A step-by-step process for preparing and casting the silicone mold is provided in S1 File.

The final device consisted of the silicone mask, HME filter, a harness and 2 elastic straps (6.4 mm x 50 cm) attached to the harness. The fully assembled respirator is shown in Figs 11–13. The final cured silicone was soft to the touch, elastic and provided an air-tight seal while worn on the face. The complete mask weighed 234 g. The cost breakdown for the first respirator model is shown in Table 2. Not included in this cost is the price of using a 3D printer such as the Prusa™MK3, which costs approximately CAD $1000. The molds can be reused to make additional units.

**Table 1. Preliminary testing of filtration systems (42 CFR Part 84 (30 L/min)).**

| System | | Seal Resistance (kPa)† | Expected (kPa) | Particles passed (count/cm³) * | Flow constant (L/min) |
|---|---|---|---|---|---|
| Mask | Filter | | | | |
| SFM | 3M 5N11–2 ply | -1.3 | < -250 | 1288 | 30 +/- 4 |
| SFM | 3M 5N11–2 ply | -1.0 | < -250 | 2040 | 30 +/- 4 |
| SFM | 3M 5N11–3 ply | -2.5 | < -250 | 3495 | 30 +/- 4 |
| Sewn Mask | Halyard H400–2 ply | -4.2 | < -250 | 271 | 30 +/- 4 |

†The expected seal resistance is negative, with < -250 kPa indicating an adequate seal.

*The no. of particles passing through in an ideal filtration system is < 20 per cubic cm in volume.

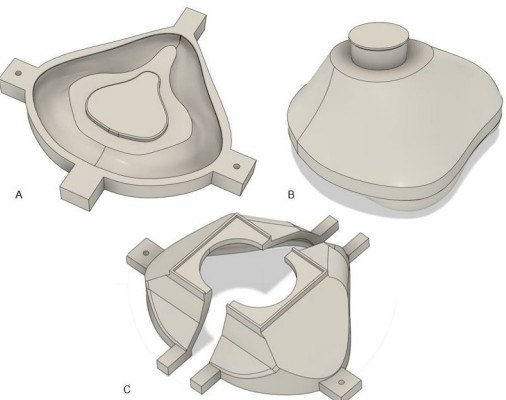

**Fig 4.** The final design of each mold part is shown; (a) base, (b) middle, (c) top cover with left and right halves.

The prototype respirator with the HME filter as shown in Fig 13 was initially tested on three of the authors (VA, WN, AM) by wearing it for 30–60 min to evaluate basic comfort and usability. Negative seal-pressure checks, by occluding the filter, and positive seal pressure checks were done to identify location of seal defects. The design was iteratively modified and tested over 8 weeks, that is the SSM was gradually remodeled with three major iterations. Once a prototype passed these qualitative assessments, preliminary quantitative fit-tests (QNFT) were performed to assess the "fit-factors" per CSA Z94.4–18 protocol using an AccuFIT™ 9000 Fit-tester Machine (Levitt-Safety, Oakville, Canada) [15]. The subject dons the respirator and is required to perform seven sequential maneuvers: normal breathing, deep breathing, turning head side-to-side, nodding, talking, bending over, and repeated normal breathing. The fit-testing device simultaneously measures the particle concentration in ambient room air as well as the concentration inside the respirator and calculates a "fit-factor" as the unitless ratio of the outside (ambient room) to inside (respirator) concentration integrals over time. For each of

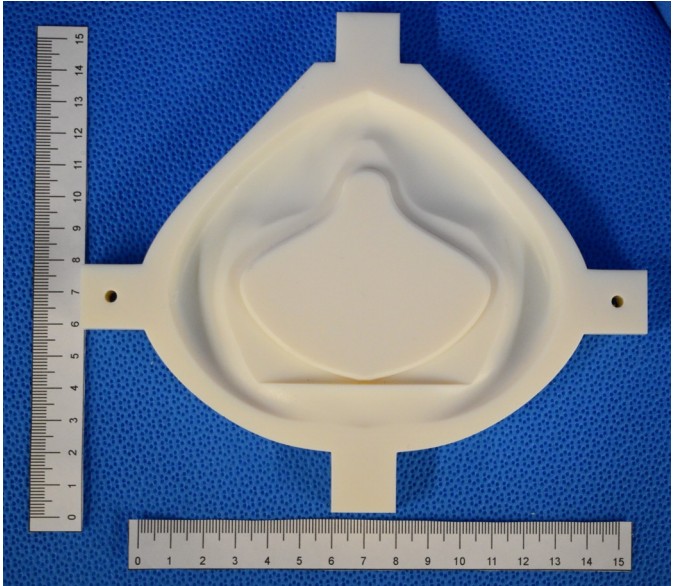

**Fig 5. 3D printed mold base is shown.**

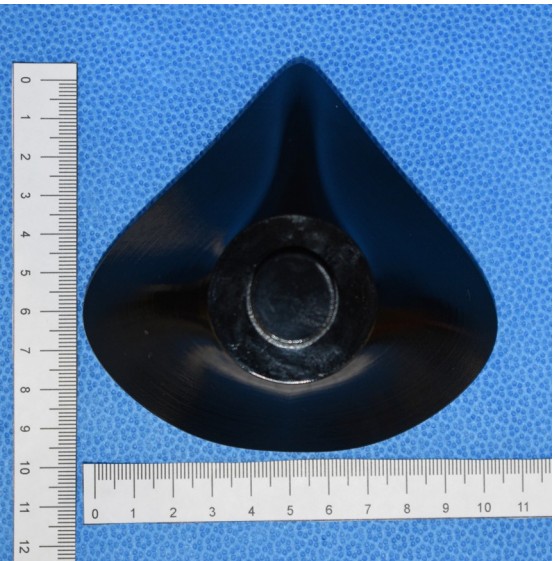

**Fig 6. 3D printed mold middle component is shown.**

the seven maneuvers, an individual fit-factor of over 100 is defined as a component pass according to CSA Z94.4–18. An overall fit-factor of over 100 is also required for an overall pass for that respirator-fitting according to CSA Z94.4–18. The overall fit-factor of the seven maneuvers is the harmonic mean of the seven runs, given by:

$$\text{Overall Fit} - \text{Factor} = \text{No. Runs (N)}/(1/ff_1 + 1/ff_2 + 1/ff_3 + \ldots + 1/ff_N),$$

where N = 7 for QNFT per CSA Z94.4–18 [16].

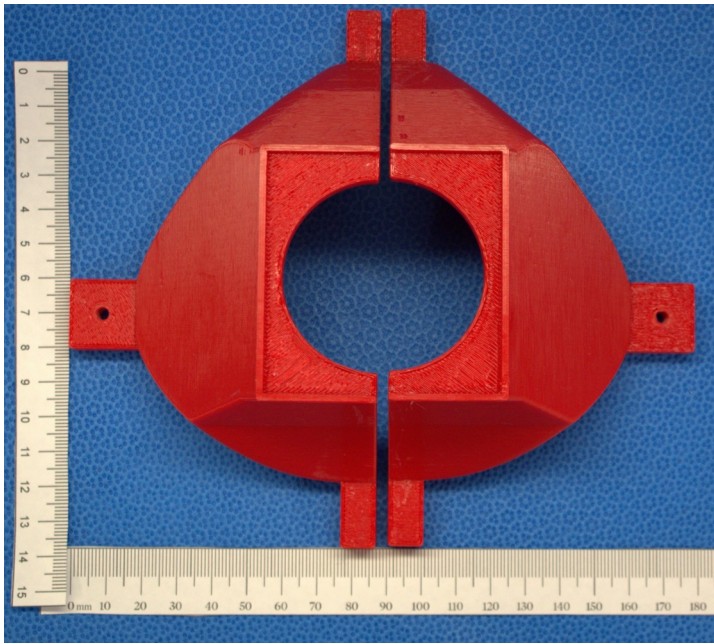

**Fig 7. 3D printed mold top cover left and right halves with a pour hole for silicone is shown.**

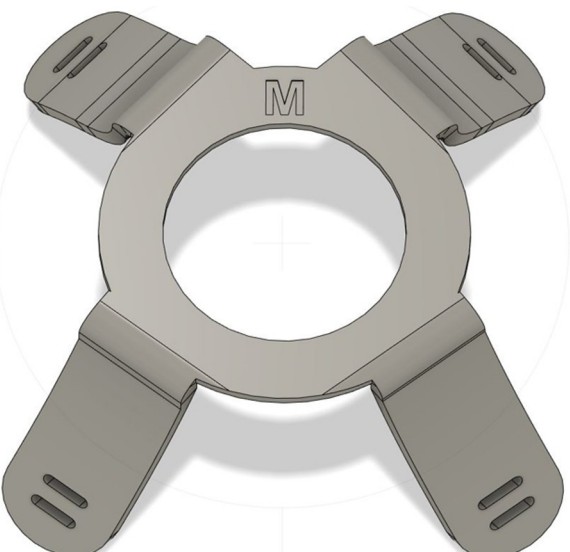

**Fig 8. Design of the final model for the harness with holes for elastic strap attachment.**

The QNFT testing study protocol was approved by the University Health Network Research Ethics Board, Toronto (REB # 20–5435.0). Initial volunteers included members of the design and testing team, and written consent was obtained for all volunteers prior to participation in the study.

## Statistical analysis

Data analyses were performed using Stata statistical software (Version 14.0, StataCorp, College Station, TX, USA), and visual data representations were created using the R package ggplot (RStudio 2020, Boston, MA, USA). The seven separate maneuvers were categorized into three stationary and four dynamic maneuvers, and the non-parametric Kruskal-Wallis equality of populations rank test was used to compare differences in the harmonic mean fit-factors between stationary and dynamic runs to identify at which points the wearer is at risk of failure. The harmonic mean was as used by the Occupational Safety and Health Administration to represent the overall fit-

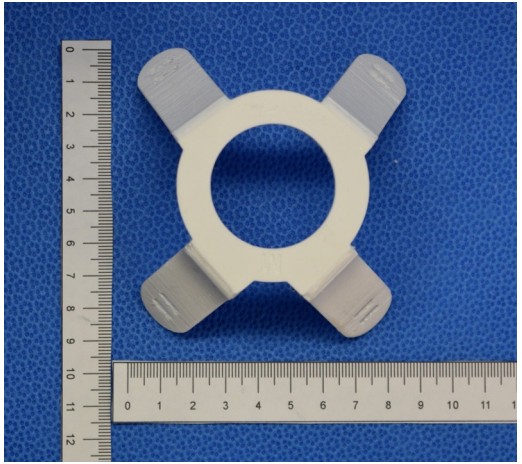

**Fig 9. 3D printed harness using PET-G plastic on the Prusa-MK3.**

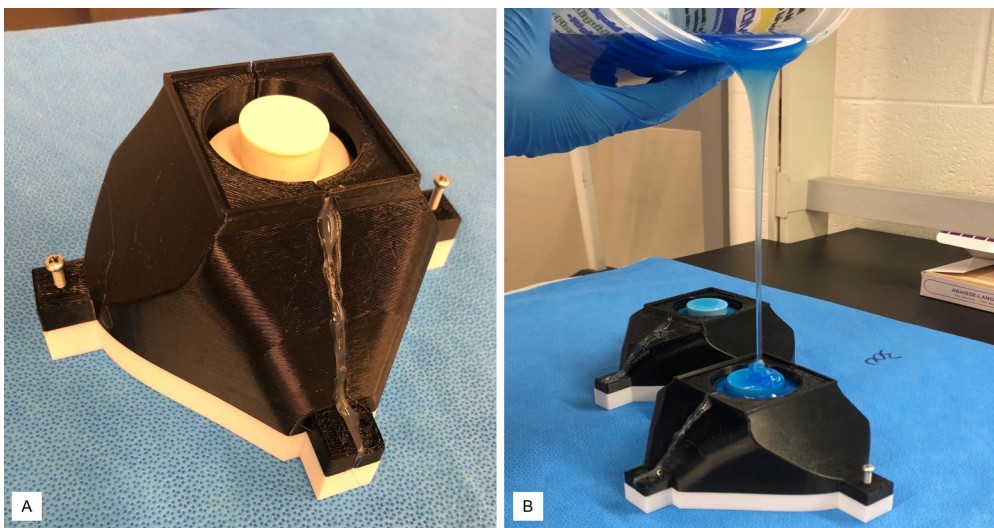

**Fig 10.** a) Assembled 4-part mold b) Silicone pouring into molds. (a) The 4-part printed mold is shown assembled using two screws. The two-halves of the cover were glued for sealing. (b). Silicone was poured from a high distance to eliminate bubble formation in the mold.

factor from individual maneuvers [16]; similarly, harmonic means of the three stationary and four dynamic maneuvers were calculated and used in statistical comparison. Median scores and logarithmic scales were used to compare large ranging numbers where appropriate.

## Results

The filtration system testing results of different mask-filter configurations are summarized in Table 1. None of the cartridge design mask (Fig 1) and filter system combinations tested

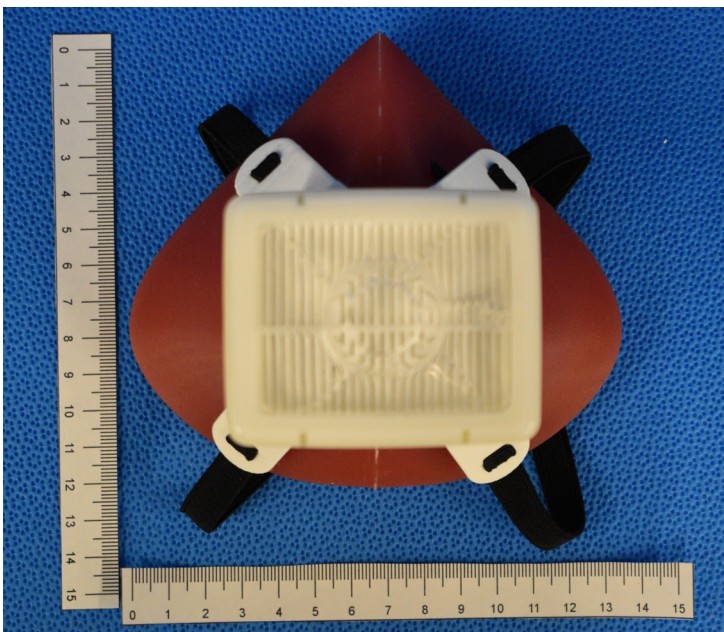

**Fig 11. Silicone mask body with harness and Intersurgical Air-Guard™ pictured.**

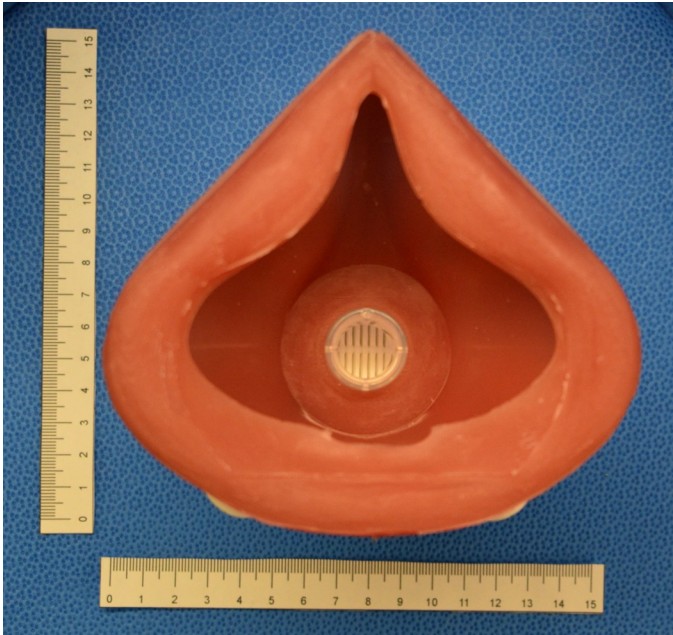

**Fig 12. Inside of the silicone mask body.**

achieved targeted filtration counts of < 20 / cm$^3$. Filtration counts ranged from ~ 1200 to 3500 / cm$^3$ for the SFM mask cartridge (Fig 1) containing 3M5N11 filters. Seal resistance ranged from –1.0 kPa for the SFM mask with 3M5N11 2-ply to– 4.2 kPa for a sewn mask composed of Halyard 400 2-ply. An effective seal pressure of a respirator would be less than– 250 kPa. The testing revealed that the resistance of the small cross-sectional area of filters led to significant flow acceleration through minor leaks around the filter, filter cartridge, and mask-seal,

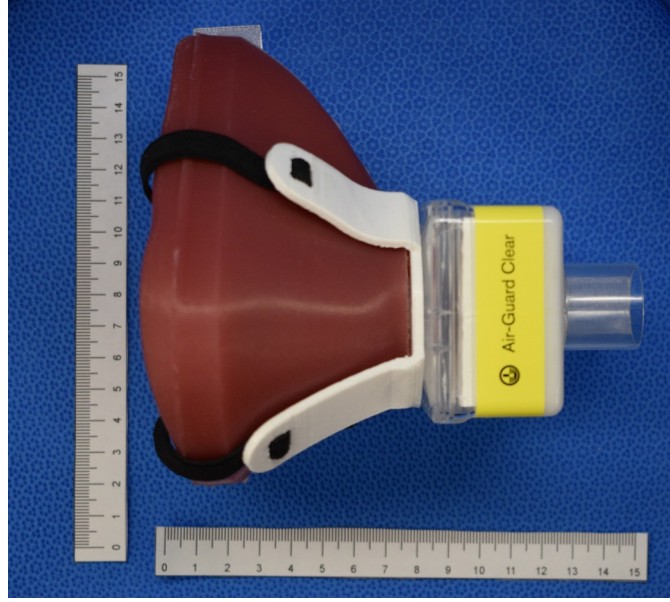

**Fig 13. Assembled SSM respirator mask.**

Table 2. Material costs and components required for a single respirator.

| Component | Material | Amount of material (g) | Cost/unit (CAD) | Development Time (h)* |
|---|---|---|---|---|
| Mold Part 1 Base | PET G filament | 94.1 | 2.4 | 8 |
| Mold Part 2 Middle | PET G filament | 122 | 3.0 | 12 |
| Mold Part 3 Cover (left and right) | PETG filament | 118 | 2.8 | 13 |
| Harness | PETG filament | 18.5 | 0.5 | 3 |
| Respirator body | Silicone | 155 | 11.6 | 1 |
| Inter-surgical Air Guard Filter | HEPA N100 rated filter | 56 | 3.6 | n/a |
| Elastic strap (64 mm wide) | Polyester and natural latex rubber | 100 | 0.3 | 0.08 |

* Development time of the mold and harness are the initial print times; when referred to the body, it is the silicone set-time. Labor time referred to in the abstract is the human hands-on time for pouring the silicone and assembly of the SSM.

resulting in poor performance. In response to these results (Table 1), we decided to use off-the-shelf medical grade respiratory HME filters with known pressure drop properties and greater than N95 filtration efficiency [17]. The focus then became adequacy of filter adapter seal and mask body seal. For this purpose, silicone or soft rubber were considered leading candidates, given their biosafe nature and industrial usage of these materials in seals of commercial respirators [18].

The resultant SSM prototype was a unibody, simple silicone mask with a single port for attachment of universal sized (22mm - 15mm od) adapters, including the respiratory filter (Fig 13). The SSM alone weighed ~155 g. There was no loss of content after overnight decontamination by soaking in 1:10 5.25% sodium hypochlorite (Chlorox™) bath and subsequent water rinsing. The circular top of the harness sat over the central port of the SSM body, allowing attachment of the filter adapter. Two elastic straps ~ 50 cm in length and 6 mm wide) were secured to the harness by looping the ends through the two harness gaps. The narrower (15 mm end) adapter port of the HME filter was then inserted into the central port. A Glia (https://glia.org) face shield was worn over the mask to check overall fitting with this additional personal protective equipment (Fig 14). The individual in Fig 14 has given written informed consent (as outlined in the PLOS consent form) to publish this case detail.

## QNFT performance during preliminary validation

The results from preliminary testing on HCWs were rapidly reported back to R&D team for analysis and refinement of design. Different respiratory HME filters were initially tried until a suitable size was determined and supply secured. DAR™ pediatric | adult mechanical filter (Medtronic, Kirkland, QC), Intersurgical Hydro-Mini™, Intersurgical Air-Guard™ (Intersurgical, Burlington, ON) were trialed at this stage. Comfort and breathability scores were also recorded out of five. Once preliminary fit-factor scores were stable, the team settled on the SSM prototype as described in this report for further subject QNFT (Fig 13).

Preliminary QNFT was performed on eight different volunteers, including three authors (VA, WN, AM). Seven runs were performed on each wearer according to CSA Z94.4–18 protocol. Fit-factors for each maneuver and each prototype iteration are presented in Table 3. The median overall fit-factor was 1662. Overall, seven out of eight tests passed. The first preliminary fit-test on the first prototype scored 108 for the overall fit-factor, but failed the runs of turning side-to-side (93), talking (83), and bending (92). The filter used was a ~ 6 x 4 x 3.5 cm$^3$ pleated HME filter Intersurgical Hydro-Mini™. The next two preliminary tests were performed on different volunteers using the second prototype, which had adjusted nasal bridge silicone padding when compared to the first prototype, as that was determined to be the source of

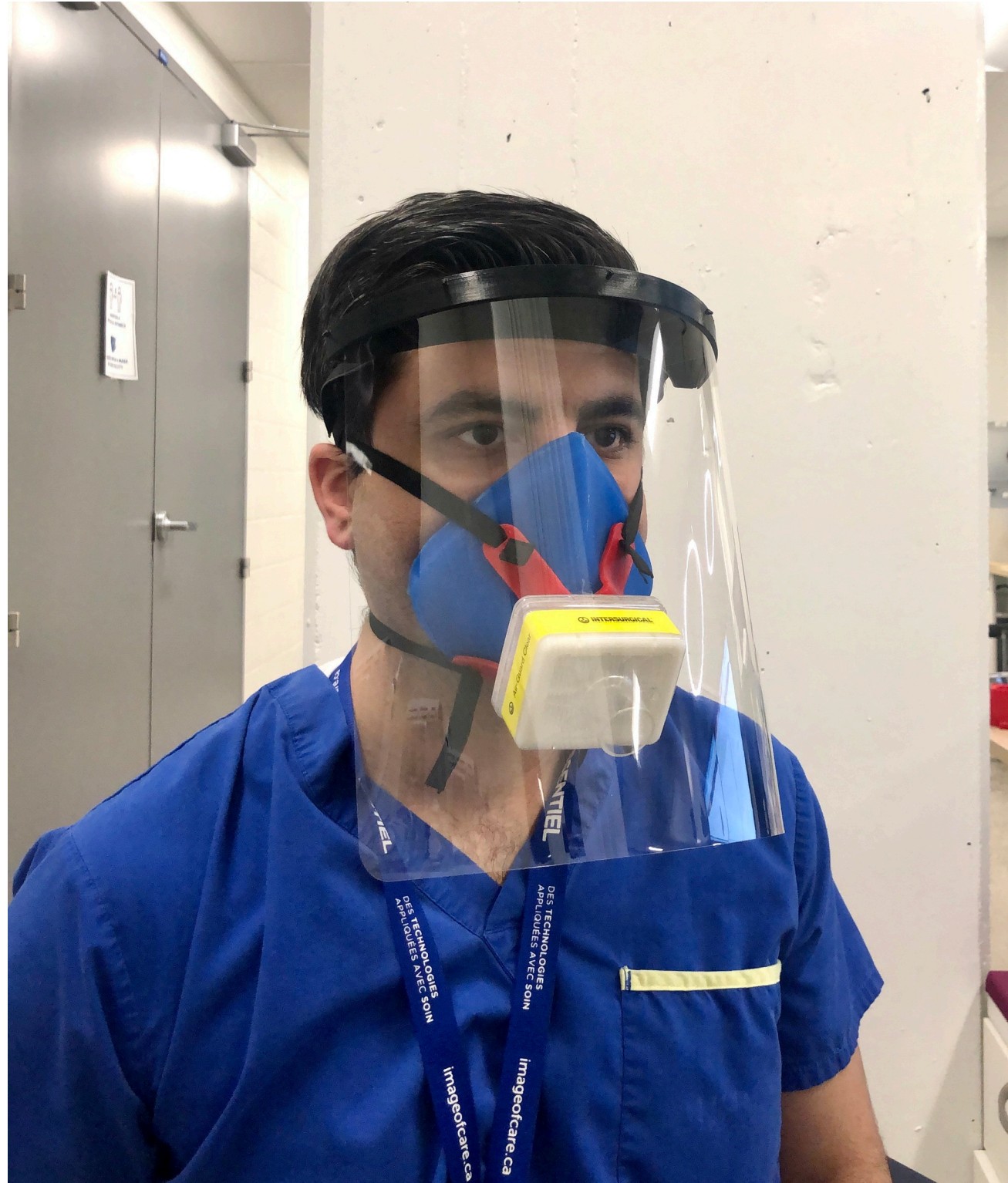

**Fig 14. The SSM respirator is worn with a face shield.**

**Table 3. Results for SSM prototype fit-tests (n = 8).**

|  | Run 1 | Run 2 | Run 3 | Run 4 | Run 5 | Run 6 | Run 7 | Overall fit-factor | Filter | SSM type |
|---|---|---|---|---|---|---|---|---|---|---|
| **1** | 196 | 118 | 93 | 127 | 83 | 92 | 104 | 108 | Hydro-Mini™ | I |
| **2** | 429 | 229 | 215 | 213 | 113 | 161 | 250 | 199 | DAR™ small | II |
| **3** | 363 | 273 | 580 | 273 | 219 | 293 | 344 | 308 | DAR™ small | II |
| **4** | >10$^6$ | 25959 | 10240 | 16958 | 4235 | 2675 | >10$^6$ | 8695 | Air-Guard™ | III |
| **5** | >10$^6$ | 10702 | >10$^6$ | 21075 | 2467 | 12724 | 18987 | 10332 | Air-Guard™ | III |
| **6** | 11144 | 7762 | 8717 | 7422 | 390 | 3131 | 8338 | 2018 | Air-Guard™ | III |
| **7** | 17210 | 7457 | 53252 | 57044 | 1133 | 61843 | >10$^6$ | 6212 | Air-Guard™ | III |
| **8** | 1556 | 1536 | 1051 | 1621 | 1369 | 1321 | 994 | 1307 | Air-Guard™ | III |

I–SSR MB-ON with nasal deficiency; II–SSM prototype with nasal bridge fix; III–SSM prototype with nasolabial pad adjusted.

minor tearing and leakage in the first prototype. Prototype 2 was paired with a DAR™ pediatric | adult mechanical filter (labeled DAR™ small in Table 3), which is of similar dimension to the previous HME filter, but from a different manufacturer. These fit-tests scored 199 and 308 overall and passed each of the seven runs.

In addition, volunteer anthropometrics, gender, BMI have been included in S1 Table.

At this stage, prototype 3 was tested on five more volunteers outside of the core team. Prototype 3 had less of a nasal vertex and more of a rounded nasal bridge rim. The nasolabial curvature was adjusted according to wearer feedback to better fit the actual fold-contour. The filter used was an Intersurgical Air-Guard™ filter, measuring ~ 6 x 8 x 3.5 cm$^3$ and almost double the effective surface exchange area of a Hydro-Mini™ filter. The lowest overall fit-factor was 1307. It was noted that in seven of the eight fit-tests, Run 5 (talking), scored the lowest for each test. In summary, seven out of the eight preliminary fit-tests passed according to the CSA protocol for N95 respirators (Fig 15). Comfort and breathability scores were 3.7 and 3.6 out of 5 respectively. The Intersurgical Air-Guard™ attached to the SSM was subjected to a NaCl permeability test regimen, 42 CFR Part 84 (at 30 L/min) [11], for filtration efficiency as a function of particle mass. The filtration efficiency was 99.7%, at a pressure drop of– 0.21 kPa.

The boxplot of the log$_{10}$ transformed composite fit-factors across all 8 volunteers are represented in Fig 16. The median composite fit-factor was 3400 (3.5 on log scale) and 1293 (3.11 on log scale) for stationary and dynamic maneuvers respectively, but this difference was not statistically significant (p = 0.2936, Kruskal-Wallis equality of populations rank test).

## Discussion

The urgent need to address depleting RPD (respiratory protective device) supplies is clear. In May 2020, Canada cut its annual order of N95 masks by 50 M from 154.4 M because of supply

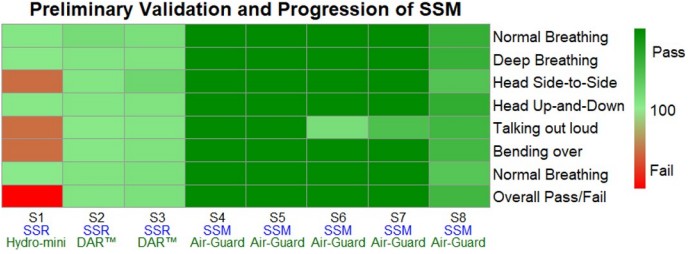

**Fig 15. Heatmap of SSM Prototype Fit-Tests (n = 8).**

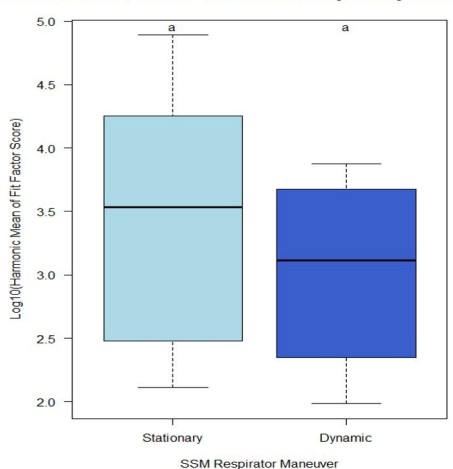

**Fig 16. Boxplot of composite fit-factors across stationary vs. dynamic maneuvers.** Boxplot of stationary and dynamic $\text{Log}_{10}$(Harmonic mean) fit-factors. The box represents the interquartile range (Q1—Q3) and the band within the box is the median. Outliers were defined as lying outside the range defined by interquartile range (IQR) +/- 1.5 IQR. The whiskers are located at the maximum and minimum values (excluding outlier denoted by "a", i.e. Subject 5).

shortage; only 1.7 M of 11.5 M N95 respirators the Canadian government has received in 2020 have passed quality control testing [19]. The continuous high demand for N95s has generated an upsurge of respirator prices. This can prevent underprivileged communities from accessing crucial PPE with adverse effect on disease transmission.

In response, researchers have attempted to develop reusable respirators to replace disposable N95s [20]. However, few novel respirators have been assessed by QNFT on HCW. This study describes the manufacture of a stopgap reusable respirator at the cost of CAD $15 for materials and 20 min of labor after initial investment in low-cost distributed manufacturing infrastructure and training of manufacturing personnel. We have used a standardized preliminary assessment of volunteer anthropometrics, user seal checks and validation by QNFT. This same methodology has been validated in our group's paper in forty subjects [21]. The authors have arrived at an investigational reusable mask design that serves as the body for N95 equivalent stopgap respirators. With further evaluation, this novel half face-piece respirator may be implemented in case of supply chain disruption during the outbreak of COVID-19, with the proviso of availability of equipment, materials, and trained personnel (see below). The wearing of facemasks is a supplementary approach to reduce disease spread in addition to physical distancing measures and hand hygiene [22].

The design simplification into two crucial challenges helped the team to firstly aim for achievable immediate gains (reusable mask body), and secondly develop future solutions to more difficult challenge of manufacturing filter materials. The particle filtration efficiency testing revealed that simple cartridge-style mask with replacement filters would not be adequate, not due to limitations of the filtering material but because of consistent difficulties in ensuring a reliable seal around the filter casing. Applying N95 grade filters of small effective surface area ($\sim 4$ x 4 cm$^2$) exaggerated seal leaks (seen in the low seal resistance measures, Table 1), and decreased overall performance (high particle count, Table 1).

With the described mask and filter combination, which utilizes known pleated-membrane HME filters with large cross-sectional areas for gas exchange, not only were the wearers able to breathe adequately, but the QNFT confirmed effective filtration efficiency or low particle counts within the respiratory chamber. In preliminary QNFT, the median fit-factor was 1662.

To give a reference range, a fit-factor of 100 approximates 99% filtration efficiency, and 1000 ~ 99.9% [23]. Once again, the SSM with pleated HME filters reached overall fit-factors of over 100, even with the smaller sized pleated-membrane filters.

There are some strengths to the use of silicone such as its inert nature, biosafety, and flexibility that gives the SSM adequate seal. In terms of comfort, the SSM prototype received positive feedback and subjective scores. Hines et al. have argued for uptake of reusable elastomeric masks in the hospital setting during a pandemic-level of demand for RPDs [1]. Any reusable material (such as thermo plastic polyurethane TPU) when shaped and formed to seal a variety of faces can be considered for local groups wishing to reproduce these models. Silicone can provide seal to tubular adapters, which can be tailored to fit various available commercial filters.

3D printing technology, originally developed as a prototyping technique, is being used increasingly for small to medium scale production in agriculture, healthcare, automotive industry, and aerospace industries [24, 25]. 3D printing is also increasingly used in low- and middle-income countries. Combined with other low-cost manufacturing techniques, such as silicone casting, such systems can provide the capacity to rapidly address immediate local needs during acute emergencies and supply chain disruptions [26]. At the time of this writing, a neighboring independent group (McMaster University, Hamilton, CA) has reproduced a workable batch of SSM within 3 weeks and is currently performing subject-testing. This stresses the need to disseminate workable and simple solutions such as the one described in this paper promptly.

Disinfection and decontamination protocols are well documented for reuse of elastomeric masks. Possible methods for decontamination cited by the CDC were vaporous hydrogen peroxide, ultraviolet germicidal irradiation, and moist heat [27]. In the hospital setting, available options include hydrogen peroxide vapor phase decontamination and autoclaving at high heat [1]. However, as per CDC instructions, solvents (e.g., acetone, ethanol) and high heat (greater than 50°C) should not be used to disinfect elastomeric respirators [20]. In our case, each component of the respirator was disinfected separately (excluding the filter containing cartridge) prior to and after each testing. The respirator body, harness and straps were disinfected using a diluted household bleach solution (5.25% sodium hypochlorite) as recommended by the CDC [20]. To prepare the solution, ~ 180 mL of household bleach was added to 3.8 L of water. The respirator components were submerged in the solution for 5 minutes, then rinsed with tap water. The components were then air-dried in a well-ventilated area prior to use. We did not reuse cartridges between volunteers; the extent of cartridge reusability (outside the scope of this study) was not investigated. The reusability of the silicone respirator body, harness and elastic straps was tested by disinfection of each component up to 50 times consecutively, with preserved integrity after 50 cycles of disinfection. The concept of a personal reusable mask could be appealing to users, and such ownership requires ease of decontamination methods such that HCWs outside of healthcare settings can easily manage their masks. The use of household bleach as recommended by the CDC was a feasible and affordable method of disinfection.

## Limitations

The current described option is reusable only to the extent of the mask body, harness, and straps. The HME filters themselves are not reusable to the same degree but their use can be prolonged. The HME filter instalment to the SSM forms both the inflow and outflow to respiration and is exposed to the wearer's droplets. Condensation and heat build-up will eventually occur over long periods of use (hours over days). HME filters will need to be dried after use, in a disaster scenario, prior to reuse. Any excess soiling to the filter casing will require a filter change. Given the relative lower cost of the filters tested compared to disposable respirators

(approximately CAD \$3 vs. CAD \$6) [17, 28], there is still an economic and environmental argument to reusing minimal number of components, and for the prolongation and decontamination of disposable parts-to-wholes. In ICU settings, typically respiratory HME filters would be changed after one day of continuous use [17]. Given the use of Air-Guard™ filters in a stopgap respirator is technically off-label, each provider group must provide their own protocol for safe duration of usage. Further research is required to evaluate filter durability and contamination (by bacterial and fungal growth sampling for example) after prolonged use.

Aesthetic and pragmatic human performance considerations are equally important including comfort and breathability. We recommend institutional field-testing to inform further modifications to maximize the acoustic quality, weight-reduction, and interaction with goggles, glasses and face-shields. The RPD is a part of the PPE armament, and must be optimized to co-fit with other protective devices. Silicone is also acoustically absorbent compared to PETG or TPU 3D printing filaments. A hybrid body that utilizes both TPU and silicone seal would increase audibility. The team has not quantified the acoustic quality of the mask-wearer at this stage of preliminary testing.

The sample size of the preliminary testing is small (n = 8) and is not enough to represent the HCW population of facial type ranges. There is no reference to the baseline performance of the respectively assigned disposable N95 respirator in the volunteers. At the time of writing, a larger comparative subject-validation trial was initiated to investigate the SSM performance [21] for a representative group of HCWs and gauge whether it was an equivalent N95 respirator.

This present report has taken a hybrid approach in addressing both the concerns of fit-testing validation for acute care clinical leadership audience, and also understanding the design and development background to an example of local-manufactured silicone-based respirator body with pleated-membrane filters. It is challenging for a specialty team to pivot sufficiently to gain expertise in all parts of this prototype development. Local leaders will need to assemble a multidisciplinary team consisting of occupational health, biomedical engineering, design, and clinical experts in order to replicate or adapt reusable respirators like the SSM. But we are publishing our experience with the expectation that the major steps outlined in this report will make reproducibility and local adoption easier.

Respiratory protection devices including half-face elastomeric respirators for use in a workplace setting including the healthcare setting must have approval from appropriate regulatory agencies, viz. NIOSH certification as per Ontario Reg 185/19 sec.10(1) under the Occupational Health and Safety Act. Therefore, The Lynn and Arnold Irwin Advanced Perioperative Imaging Lab (APIL) and its affiliates do not endorse the use of the stopgap respirator described above until such time as additional testing and regulatory approval have been obtained. This half-face respirator as described is an investigational device under development and has not passed all relevant tests for safety and effectiveness and does not currently meet all regulatory requirements for respirators in Canada.

## Conclusion

The challenge to PPE and respiratory protective device supply will continue given the state of the COVID-19 pandemic. Respirators of N95 grade will be in demand in the foreseeable future. The production by distributed manufacturing has the advantage of access, low-cost, reusability, and reliability in supply. We have described the process of arriving at a reusable N95 grade respirator using a simple reiterative design and production process, off-the-shelf HME filters, 3D printing and silicone casting. The SSM prototype is only one of many options that can potentially be reproduced and tailored to meet the local and regional HCW respirator

needs. Regulatory requirements must be met before RPDs are to be used in the non-emergent setting. Lastly, the need for larger subject validation and field-testing of such reusable respirators is taxing but worthwhile given the stakes and benefits of prolonged respirator reusability in a protracted pandemic course.

## Supporting information

**S1 Table. Demographics and anthropometric characteristics of participants.**
(DOCX)

**S1 File. 3D Print settings and silicone casting process.**
(DOCX)

**S2 File. De-identified dataset N8.**
(XLSX)

## Acknowledgments

The authors sincerely thank the UHN Department of Anesthesiology, The Lynn and Arnold Irwin Advanced Perioperative Imaging Lab, and The Toronto General and Western Hospital Foundation for their academic support. Thanks to Christian Petropolis (University of Manitoba), Thomas Looi (SickKids CIGITI), Brandon Peel (SickKids CIGITI), and Matt Ratto (Faculty of Information, University of Toronto UofT) for their concept, design, and production input. We are grateful for the *pro bono* printing provided by Steve Cory (Objex Unlimited, Mississauga, Canada), James Garel-Jones (Vertigro, Toronto), and Brian Read (Coburg, Canada). Thanks to James Scott et al. at the Gage Lab, Occupational Environmental Health, University of Toronto for their filtration testing. Intersurgical (Michael Hayden, Burlington, Canada) was most generous in donation of respiratory filters. The Techna Institute (Toronto) provided vital laboratory space amidst the UHN COVID response. Thanks to Chris Murray, Lakehead University (Orillia, Canada), for filter material testing, and Andre Lafreniere (Thunder Bay, Canada) for design input. The Toronto Emergent Device Accelerator platform, 3D-PPE-GTHA, Kingston PPE, NOSM PPE, and USASK PPE groups (Canada) provided important student and faculty networks and bolstered the design process. Thanks to Alana Bernick (UofT) for proof-reading from an occupational health perspective. Final thanks belong to Andrew Syrett (MD, McMaster University), Nasa Nguyen and Natasha Valenton (BAScEng, University of Toronto), without whose design and creativity the respirator development could not have started and progressed to a stopgap solution.

## Author Contributions

**Conceptualization:** Vahid Anwari, William C. K. Ng, Azad Mashari.

**Data curation:** Vahid Anwari, William C. K. Ng, Arnaud Romeo Mbadjeu Hondjeu, Zixuan Xiao, Jessica Trac.

**Formal analysis:** William C. K. Ng, Arnaud Romeo Mbadjeu Hondjeu, Zixuan Xiao.

**Funding acquisition:** Azad Mashari.

**Investigation:** Vahid Anwari, William C. K. Ng.

**Methodology:** Vahid Anwari, William C. K. Ng, Joshua Hiansen.

**Project administration:** Vahid Anwari, William C. K. Ng.

**Resources:** Joshua Hiansen, Azad Mashari.

**Software:** Zixuan Xiao.

**Supervision:** William C. K. Ng, Azad Mashari.

**Validation:** William C. K. Ng, Arnaud Romeo Mbadjeu Hondjeu.

**Visualization:** Vahid Anwari, William C. K. Ng, Zixuan Xiao, Edem Afenu, Kate Kazlovich.

**Writing – original draft:** Vahid Anwari, William C. K. Ng, Arnaud Romeo Mbadjeu Hondjeu.

**Writing – review & editing:** Vahid Anwari, William C. K. Ng, Arnaud Romeo Mbadjeu Hondjeu, Edem Afenu, Jessica Trac, Kate Kazlovich, Azad Mashari.

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
