## [Decision Letter · Decision Letter 0]

11 Dec 2020

PONE-D-20-21977

Development, manufacturing, and preliminary validation of a reusable half-face respirator during the COVID-19 pandemic

PLOS ONE

Dear Dr. William C. K. Ng,

Thank you for submitting your manuscript to PLOS ONE. After careful consideration, we feel that it has merit but does not fully meet PLOS ONE’s publication criteria as it currently stands. Therefore, we invite you to submit a revised version of the manuscript that addresses the points raised during the review process.

We look forward to receiving your revised manuscript.

Kind regards,

Tommaso Lomonaco, Ph.D

Academic Editor

PLOS ONE

Journal Requirements:

2. Please clarify in your Ethics statement and Methods section the full name of the ethics committee that approved the study, along with the approval number. Please also clarify how the the participants gave consent.

3.Thank you for stating the following financial disclosure:

 [No].

4. Please ensure that you refer to Figure 3 in your text as, if accepted, production will need this reference to link the reader to the figure.

5.We note that Figure [9] includes an image of a patient / participant in the study. 

Please respond by return e-mail with an amended manuscript. We can upload this to your submission on your behalf.

If you are unable to obtain consent from the subject of the photograph, please either instruct us to remove the figure or supply a replacement figure by return e-mail for which you hold the relevant copyright permissions and subject consents. In some cases, you may need to specify in the text that the image used in the figure is not the original image used in the study, but a similar image used for illustrative purposes only. We can make any changes on your behalf.

6. Please ensure that your related published work https://journals.plos.org/plosone/article?id=10.1371/journal.pone.0242304 is referenced within the manuscript.

Additional Editor Comments:

Dear Authors,

please find attached the reviewers' comments. Please answer in details all the questions raised during the revision process.

Best regards.

Tommaso Lomonaco

Reviewers' comments:

Reviewer's Responses to Questions

**Comments to the Author**

1. Is the manuscript technically sound, and do the data support the conclusions?

Reviewer #1: Yes

Reviewer #2: Partly

2. Has the statistical analysis been performed appropriately and rigorously? 

Reviewer #1: Yes

Reviewer #2: No

3. Have the authors made all data underlying the findings in their manuscript fully available?

Reviewer #1: Yes

Reviewer #2: No

4. Is the manuscript presented in an intelligible fashion and written in standard English?

Reviewer #1: No

Reviewer #2: Yes

5. Review Comments to the Author

Reviewer #1: This is a very timely and well-written manuscript. Essentially, the research team has developed a reusable silicone half-facepiece respirator that can be used when there are supply challenges with existing respiratory protection used in healthcare e.g. N95 respirator for airborne diseases including COVID-19.

One of the major shortcomings of the study is that the authors only conducted testing of the respirator on eight different volunteers. This is a small number to begin with and the authors have acknowledged that this is a study limitation. However, in order to strengthen their study, they should, in my opinion, include the gender, weight and anthropometric dimensions of each volunteer’s face to ascertain if they are typical of respirator users such as the article by Zhuang and Bradtmiller (2005) Head-and-face anthropometric survey of US respirator users. Journal of occupational and environmental hygiene 2(11):567-76. This is important because commercially-available respirators come in different shapes and sizes and it would behoove the research team to include this information so that the reader is confident that the novel respirator is suitable for his/her workforce.

Upon review of the research team, there does not appear to be anyone with a formal background in occupational hygiene or occupational health and safety. This has resulted in some obscure terminology and/or errors. For instance, in the abstract, the team uses the phrase “user pressure test” which is commonly referred to as “user seal check” among the occupational health and safety community.. Also, on line 126 under “Methods”, NIOSH is the acronym for the “National Institute for Occupational Health and Safety” (the authors used “of” instead).

Lastly, but most importantly, I am not sure if the research team is aware that any respiratory protection selected for use in a workplace must meet approval requirements e.g. NIOSH-certified as per Ont. Reg 185/19 sec.10(1) under the Occupational Health and Safety Act (this legislation was selected because the authors are based in Ontario, Canada). Given that this is a legal requirement, I believe that the authors should explicitly have a statement regarding this matter in their manuscript (such as in the abstract as well as in the Discussion). Despite the promise of their novel respirator, it cannot technically be used in a workplace without the device being certified first.

Given the above, I therefore urge the team to consider seeking an expert in occupational health and safety to ensure that terminology and statements within their manuscript is correct.

In the Methods section, there is no mention about institutional ethics. Since human subjects were involved, it is assumed that research ethics is required. The authors should provide clarity on this.

In the Methods section. No justification or basis for conducting differences in the median harmonic mean fit-factor is provided i.e. there is no reference cited for this test e.g. CSA Z94 and/or another published study. The authors should provide clarity on this.

In the Discussion section, the authors stress the importance of disinfection and decontamination of reusable elastomeric masks. However, the authors did not actually test the feasibility of disinfection and reuse of their novel respirator (not to mention how many times the mask mold can be disinfected before the integrity becomes questionable). This should be mentioned as a limitation to their study because the ability to reuse a protective device is incredibly important.

In Figure 9, the worker is wearing a face shield in addition to the novel respirator. Have the authors tested additional protective equipment such as goggles/safety glasses as well as a combination of face shield and goggles/safety glasses with the respirator? The authors should provide clarity on this.

In the Methods section, the sub-section related to “Model 2” is extremely technical. I am not sure who the intended audience is but it would be really helpful for a reader if this could be simplified somewhat so that they will be able to replicate the respirator within their own organization. Also, please define all acronyms the first time they are used e.g. PLA and PETG.

Reviewer #2: This is a very topical and useful paper in this pandemic and for the future. So it needs publication. It will add prestige to the journal, I suggest mandatory revisions and re-review:

1. Recent overarching specific literature is not covered and related to this work, see a very original paper on masks related to this pandemic by Ahmed et al in Med Devices & Sensors. Also, a very case specific paper very relevant to this work by Alenezi et al. in BioDesign & Manufacturing. Discuss your work in relation to these.

2. There has to be more evaluation in terms of discussion and statistics on the manufacturing of this device. Is it cost-effective?

3. Particles passed (Table 1) re-check and give in whole numbers, more stats required.

4.Figs 4-7 and may be 8 too come in quick succession, more discussion required here.

5." Fit-factor of over 100 is a pass." - ??? Explain more. Units are all over the place, USE SI units only in entire manuscript.

6. "3D printing is now commonly available to community groups and almost all countries

including middle-income countries. The use of affordable, low-cost 3D printers will allow users

to rapidly print molds. " - needs more evidence and discussion as I said before (under 2), manufacturing is a key factor.

7. Scales needed in all figures.

6. PLOS authors have the option to publish the peer review history of their article (what does this mean?). If published, this will include your full peer review and any attached files.

Reviewer #1: No

Reviewer #2: No

---

## [Author Response · Author response to Decision Letter 0]

7 Jan 2021

This is a copy of the new cover letter to the Editor and the Reviewers.

"Dr. Tommaso Lomonaco, Ph.D

Academic Editor

PLOS ONE 

Toronto, Dec 18th, 2020 

RE: Response to Reviews of “Development, manufacturing, and preliminary validation of a reusable half-face respirator during the COVID-19 pandemic”

ID: PONE-D-20-21977

Dear Dr. Tommaso Lomonaco, Reviewer1, and Reviewer 2,

Thank you all for your time in reviewing the submission. The editorial and review comments are invaluable and are addressed sequentially, as outlined in the decision letter “PONE-D-20-21977 Decision Major Revision 2020 Dec 11” 

1. The MS and supporting files has been edited and renamed to meet PLOS ONE requirements.

2. The full name of the ethics committee which approved the study has been included in the Methods section (University Health Network, Research Ethics Board). The approval number has also been included (REB# 20-5435.0). Written consent was obtained for all volunteers in the study. This has been added in the Methods section as well. 

3. Regarding funding and salary:

a, b, d. The authors received no specific funding for this work.

c. The authors received no funding nor salary for this work.

Thank you for changing the online submission form on our behalf.

4. Figure 3 has been referred to in the MS text.

5. The image in Figure [9] is of the first author (Vahid Anwari) listed in MS. The author has downloaded the Consent Form, has signed and filed the case notes. 

The following text has been added to the Methods section, after Figure 9, to comply with PLOS ONE guidelines: “The individual in Figure 9 has given written informed consent (as outlined in the PLOS consent form) to publish this case detail.” 

6. The authors’ related published work has been referenced in the MS test: (https://journals.plos.org/plosone/articleid=10.1371/journal.pone.0242304).

7. We have added the raw testing dataset into a supporting file.

8. We have added more statistical explanation as suggested by the reviewers.

 

Response to Reviewer 1 and 2’s comments: 

Thank you for the detailed comments, they were instructive, provided valid criticisms and pointed out areas of improvements to the MS: 

Reviewer #1: “This is a very timely and well-written manuscript. Essentially, the research team has developed a reusable silicone half-facepiece respirator that can be used when there are supply challenges with existing respiratory protection used in healthcare e.g., N95 respirator for airborne diseases including COVID-19.”

• Thank you for the encouragement.

“One of the major shortcomings of the study is that the authors only conducted testing of the respirator on eight different volunteers. This is a small number to begin with and the authors have acknowledged that this is a study limitation. However, in order to strengthen their study, they should, in my opinion, include the gender, weight and anthropometric dimensions of each volunteer’s face to ascertain if they are typical of respirator users such as the article by Zhuang and Bradtmiller (2005) Head-and-face anthropometric survey of US respirator users. Journal of occupational and environmental hygiene 2(11):567-76. This is important because commercially-available respirators come in different shapes and sizes and it would behoove the research team to include this information so that the reader is confident that the novel respirator is suitable for his/her workforce.”

• We have added the gender, weight and anthropometric dimensions of each volunteer in a table format and referenced it within the MS text. This is a welcomed comment.

“Upon review of the research team, there does not appear to be anyone with a formal background in occupational hygiene or occupational health and safety. This has resulted in some obscure terminology and/or errors. For instance, in the abstract, the team uses the phrase “user pressure test” which is commonly referred to as “user seal check” among the occupational health and safety community. Also, on line 126 under “Methods”, NIOSH is the acronym for the “National Institute for Occupational Health and Safety” (the authors used “of” instead).”

• The NIOSH’s expanded title has been rectified.

• All references to “user seal check” has been rephrased as such.

Lastly, but most importantly, I am not sure if the research team is aware that any respiratory protection selected for use in a workplace must meet approval requirements e.g. NIOSH-certified as per Ont. Reg 185/19 sec.10(1) under the Occupational Health and Safety Act (this legislation was selected because the authors are based in Ontario, Canada). Given that this is a legal requirement, I believe that the authors should explicitly have a statement regarding this matter in their manuscript (such as in the abstract as well as in the Discussion). Despite the promise of their novel respirator, it cannot technically be used in a workplace without the device being certified first.

• The following issue is now extensively addressed in the Limitations section, with specific reference to the Ontario Reg 185/19 sec.10(1) under the Occupational Health and Safety Act. The need for further testing in order to meet regulatory requirement and the non-endorsement of the described investigational device is stated.

• The following text has been added in the abstract’s conclusion: ‘The product described above is an investigational device and does not currently meet regulatory requirements for a respirator. Therefore, the authors and/or any affiliates do not make any endorsements for the use of products described above.’

“Given the above, I therefore urge the team to consider seeking an expert in occupational health and safety to ensure that terminology and statements within their manuscript is correct.”

• An OH and Biomedical Colleague has been invited to read the final submission for this express purpose.

“In the Methods section, there is no mention about institutional ethics. Since human subjects were involved, it is assumed that research ethics is required. The authors should provide clarity on this.”

• Our apologies in not clarifying institutional ethics approval, thank you for pointing out. The full name of the ethics committee which approved the study has been included in the Methods section (University Health Network, Research Ethics Board). The approval number has also been included (REB# 20-5435.0). Written consent was obtained for all volunteers in the study. This has been added in the Methods section as well.

“In the Methods section. No justification or basis for conducting differences in the median harmonic mean fit-factor is provided i.e. there is no reference cited for this test e.g. CSA Z94 and/or another published study. The authors should provide clarity on this.”

• Firstly, the formula for the overall fit-factor is a harmonic mean, given by the formula Overall Fit-Factor = No. Runs (N) / (1/ff1 + 1/ff2 + 1/ff3 + … +1/ffN), 

where N = 7 for QNFT per CSA Z94.4-18 and OHSA of the USA (see Reference 15). 

• Looking at the range of overall fit-factors (108 to 10332), we decided to use a median for such skewed raw data. Log transformation has been used also where appropriate.

“In the Discussion section, the authors stress the importance of disinfection and decontamination of reusable elastomeric masks. However, the authors did not actually test the feasibility of disinfection and reuse of their novel respirator (not to mention how many times the mask mold can be disinfected before the integrity becomes questionable). This should be mentioned as a limitation to their study because the ability to reuse a protective device is incredibly important.”

• The disinfection and cleaning protocol for the respirator has been added in the Discussion. 

“In Figure 9, the worker is wearing a face shield in addition to the novel respirator. Have the authors tested additional protective equipment such as goggles/safety glasses as well as a combination of face shield and goggles/safety glasses with the respirator? The authors should provide clarity on this.”

• Yes partially: indeed in preparation for the Figure as well in practice runs, we have worn face-shields on top of the mask prototype. We have not applied goggles in every test run on top of the mask. But we have been able to apply the volunteer’s own spectacles as they would normally wear on top of the mask. This has been clarified in the discussion.

“In the Methods section, the sub-section related to “Model 2” is extremely technical. I am not sure who the intended audience is but it would be really helpful for a reader if this could be simplified somewhat so that they will be able to replicate the respirator within their own organization. Also, please define all acronyms the first time they are used e.g. PLA and PETG.”

• We have simplified “Model 2”’s technical subsection and moved the detailed version into a supporting file.

• We have expanded the TLA PLA and PETG in several places in the MS and the supporting file.

Response to Reviewer 2’s comments: 

Reviewer #2: “This is a very topical and useful paper in this pandemic and for the future. So, it needs publication. It will add prestige to the journal, I suggest mandatory revisions and re-review:”

1. “Recent overarching specific literature is not covered and related to this work, see a very original paper on masks related to this pandemic by Ahmed et al in Med Devices & Sensors. Also, a very case specific paper very relevant to this work by Alenezi et al. in BioDesign & Manufacturing. Discuss your work in relation to these.”

• Thank you for these two references by Ahmed et al. and Alenezi et al. We have added a discussion of our work in relation to these two original papers and will add these to a 3rd follow-up paper to this one. (our group’s 2nd paper has been referenced and published in PLOS ONE Nov 2020)

2. “There has to be more evaluation in terms of discussion and statistics on the manufacturing of this device. Is it cost-effective?”

• Cost-effectiveness in terms of dollars per unit and hours of manual labor per unit has been added in the discussion section.

3. “Particles passed (Table 1) re-check and give in whole numbers, more stats required.”

• The “particles passed” column has been rechecked and given in whole numbers; but no additional relevant analysis added upon discussion with our statistical personnel for this column.

4. “Figs 4-7 and may be 8 too come in quick succession, more discussion required here.”

• We have now spread-out references to Fig 4-7 with discussion sentences placed in between.

5. “" Fit-factor of over 100 is a pass." - ??? Explain more. Units are all over the place, USE SI units only in entire manuscript.”

• Thank you for the suggestions. We have expanded the sentence on “fit-factor” to include each individual maneuver’s fit-factor and the overall fit-factor. Units are now in SI: where m, cm, mm, um, and nm are appropriate, we have used that for distance; inches of water are now expressed in kPa for pressure (we know that this breaks tradition with the pressure units in our two NIOSH references and some other referenced articles, but it is now internally consistent and easily translatable by the thorough reader).

6. “"3D printing is now commonly available to community groups and almost all countries including middle-income countries. The use of affordable, low-cost 3D printers will allow users to rapidly print molds. " - needs more evidence and discussion as I said before (under 2), manufacturing is a key factor.”

• We accept the comment that the two sentences require more evidence, and therefore we have rephrased the comment into an admission of need for further evidence of accessibility of equipment and manufacturing feasibility in middle-income countries.

7. “Scales needed in all figures.”

• We have added scales to figures.

We hope the major revision to the MS as indicated above has made this submission acceptable to PLOS, in line with the Journal’s rigorous academic standards. Thank you all once again for the reviews, comments and suggestions.

Truly,

William C. K. Ng

MBBS MMed FANZCA FRCPC

Anesthesia and Pain Management, Toronto General Hospital

TGH Research Institute & UHN Advanced Perioperative Imaging Lab

Pediatric Cardiac Anesthesia, SickKids Hospital

Faculty Dept. of Anesthesiology, University of Toronto

M 1.857.330.7399

E William.Ng@uhn.ca or William.Ng@sickkids.ca

Soli Deo Gloria"

---

## [Decision Letter · Decision Letter 1]

27 Jan 2021

PONE-D-20-21977R1

Development, manufacturing, and preliminary validation of a reusable half-face respirator during the COVID-19 pandemic

PLOS ONE

Dear Dr. William C. K. Ng,

Thank you for submitting your manuscript to PLOS ONE. After careful consideration, we feel that it has merit but does not fully meet PLOS ONE’s publication criteria as it currently stands. Therefore, we invite you to submit a revised version of the manuscript that addresses the points raised during the review process.

We look forward to receiving your revised manuscript.

Kind regards,

Tommaso Lomonaco, Ph.D

Academic Editor

PLOS ONE

Additional Editor Comments (if provided):

Dear authors, all the questions raised during the revision process has been answered. Anyway, the article requires additional minor revisions.

Regards,

Tommaso Lomonaco

Reviewers' comments:

Reviewer's Responses to Questions

**Comments to the Author**

1. If the authors have adequately addressed your comments raised in a previous round of review and you feel that this manuscript is now acceptable for publication, you may indicate that here to bypass the “Comments to the Author” section, enter your conflict of interest statement in the “Confidential to Editor” section, and submit your "Accept" recommendation.

Reviewer #1: (No Response)

Reviewer #2: All comments have been addressed

2. Is the manuscript technically sound, and do the data support the conclusions?

Reviewer #1: Yes

Reviewer #2: Yes

3. Has the statistical analysis been performed appropriately and rigorously? 

Reviewer #1: Yes

Reviewer #2: Yes

4. Have the authors made all data underlying the findings in their manuscript fully available?

Reviewer #1: Yes

Reviewer #2: Yes

5. Is the manuscript presented in an intelligible fashion and written in standard English?

Reviewer #1: Yes

Reviewer #2: Yes

6. Review Comments to the Author

Reviewer #1: The authors have done an admirable job responding to the previous comments. However, there are still some matters that need to be addressed - see below. The other point that I want to bring to the attention of the authors is who, exactly, is the intended audience? I ask because there appear to be various disciplines addressed in the manuscript and, in my opinion, it would be a challenge for one discipline to understand and recreate all elements in the manuscript for their practice. In other words, it appears that someone knowledgeable in 3D printing wrote part of the ms, someone with occupational health background wrote other segments, and the statistically analyses was written by someone else entirely. Personally, this made it very hard for me to follow as a reviewer. I bring this forward because, since this manuscript is meant to have practical applications, it might be beneficial to explain which departments from an acute care facility need to be involved to produce these devices. (Perhaps this could be an entirely separate manuscript altogether?)

Page 4, line 75. Grammatical error. Word should be “affecting” not “effecting”

Page 4, lines 87-88. The statement is technically incorrect. N95 respirators are 95% effective in filtering aerosols around 0.3 μm (which is the most penetrating particle). This is not explicitly stated in the referenced Brosseau CDC blog - one needs to “read between the lines” but this is the normally accepted definition. See https://www.sciencedirect.com/science/article/pii/S2452199X20301481 and https://www.cdc.gov/niosh/docs/96-101/default.html as examples.

Page 5, line 94. Please define “SSM”.

Page 6, line 108. Grammatical error. Should be “designed” not “designs”.

Page 6, lines 123 - 124. Please provide a reference for the statement: “General purpose health care respirators must achieve 95% filtration of particles down to 10 nm”.

Page 9, Table 2. The authors have mentioned that producing the new device is cost effective and does not require much in terms of labour. However, in terms of development time in Table 2, it appears to be nearly 40 hrs per single respirator! If the primary objective of undertaking this study is to off-set supply shortages of PPE during a pandemic, then the development time needs to be discussed as this is a potential limiting factor especially in an acute care hospital where potentially hundreds of front-line care workers might require PPE.

Page 10, lines 192-193. I am unclear what this sentence means. “The design was iteratively modified and rested over 8 weeks”. Please clarify and/or elaborate…

Page 10, line 198. It should be “The Fit-testing device” not the “The Fit-tester device”.

Page 14, lines 271-274. Paragraph should be relocated to be the first paragraph after the subheading “QNFT performance during preliminary validation”.

Page 14, lines 279-280. Although I appreciate that the authors included anthropometric data, they really ought to compare their test subjects with anthropometric data of typical respirator users as per Zhuang and Bradtmiller (2005) Head-and-face anthropometric survey of US respirator users. Journal of occupational and environmental hygiene 2(11):567-76. This comparison is important to allow the reader to ascertain if the novel respirator would be suitable for his/her workforce.

Figure 03. Text is not clear in the red boxes.

S1 Table. What does “NIOSH Panel” refer to?

Reviewer #2: Sensible responses and revisions made. So, accept. The authors have given reasons for the changes. For easiness of review it would have been better if changes were highlighted in red on R1.

7. PLOS authors have the option to publish the peer review history of their article (what does this mean?). If published, this will include your full peer review and any attached files.

Reviewer #1: No

Reviewer #2: No

---

## [Author Response · Author response to Decision Letter 1]

29 Jan 2021

Dr. Tommaso Lomonaco, Ph.D

Academic Editor

PLOS ONE 

Toronto, Jan 28th, 2020 

RE: Response to 2nd Review of “Development, manufacturing, and preliminary validation of a reusable half-face respirator during the COVID-19 pandemic”

ID: PONE-D-20-21977R1

Dear Dr. Tommaso Lomonaco, Reviewer1, and Reviewer 2,

Thank you all for your time in reviewing the re-submission. The additional comments are now addressed sequentially, as outlined in the decision letter “PONE-D-20-21977 Decision Minor Revision 2021 Jan 27” 

---

Dear Reviewer #1: 

“The authors have done an admirable job responding to the previous comments. However, there are still some matters that need to be addressed - see below. The other point that I want to bring to the attention of the authors is who, exactly, is the intended audience? I ask because there appear to be various disciplines addressed in the manuscript and, in my opinion, it would be a challenge for one discipline to understand and recreate all elements in the manuscript for their practice. In other words, it appears that someone knowledgeable in 3D printing wrote part of the [MS], someone with occupational health background wrote other segments, and the statistically analyses was written by someone else entirely. Personally, this made it very hard for me to follow as a reviewer. I bring this forward because, since this manuscript is meant to have practical applications, it might be beneficial to explain which departments from an acute care facility need to be involved to produce these devices. (Perhaps this could be an entirely separate manuscript altogether?)”

- We agree that this is a major challenge for one discipline to understand and recreate all elements in the manuscript for local use. Without resorting to having a separate MS for the disparate 3D-printing elements alone, we want to keep the acute care audience’s primary focus on the validation and reiterative process of the prototype design, and secondarily allow the same non-expert audience to glean the major steps of the development and manufacturing background. This way, the recipient leader(s) can gauge what team members (such as biomedical engineering and design) and production hurdles would need to be overcome to successfully replicate this method, adapted to the local needs. We have added a standalone paragraph in the limitation section to draw attention to this hybrid aim. Page 21, lines 413-421.

“Page 4, line 75. Grammatical error. Word should be “affecting” not “effecting””

- Thanks, we have corrected this. Page 5, line 75.

“Page 4, lines 87-88. The statement is technically incorrect. N95 respirators are 95% effective in filtering aerosols around 0.3 μm (which is the most penetrating particle). This is not explicitly stated in the referenced Brosseau CDC blog - one needs to “read between the lines” but this is the normally accepted definition. See https://www.sciencedirect.com/science/article/pii/S2452199X20301481 and https://www.cdc.gov/niosh/docs/96-101/default.html as examples.”

- Thank you. We have rephrased the statement to be more accurate as suggested and added the first suggested reference by Tcharkhtchi et al. to point to this technical correction. Page 5, lines 87-88.

“Page 5, line 94. Please define “SSM”.”

- SSM is an assigned name of the respirator device, the derivation is now spelled out in the abstract. Page 6, line 94.

“Page 6, line 108. Grammatical error. Should be “designed” not “designs”.”

- This has been left as “design” because we are referring to the design object by Chris Richburg, not the action of Chris Richburg. Page 7, line 106.

“Page 6, lines 123 - 124. Please provide a reference for the statement: “General purpose health care respirators must achieve 95% filtration of particles down to 10 nm”.”

- In view of the suggestion to Page 4, lines 87-88, we have rephrased this to be consistent with the added reference to Tcharkhtchi et al.’s article and referenced to the same. Page 8, line 124.

“Page 9, Table 2. The authors have mentioned that producing the new device is cost effective and does not require much in terms of labour. However, in terms of development time in Table 2, it appears to be nearly 40 hrs per single respirator! If the primary objective of undertaking this study is to off-set supply shortages of PPE during a pandemic, then the development time needs to be discussed as this is a potential limiting factor especially in an acute care hospital where potentially hundreds of front-line care workers might require PPE.”

- Thanks for pointing this out. We have clarified development times and manual labor input times with the following comment to Table 2: “Development time of the mold and harness are the initial print times; when referred to the body, it is the silicone set-time. Labor time referred to in the abstract is the human hands-on time for pouring the silicone into the molds and assembly of the SSM.”

“Page 10, lines 192-193. I am unclear what this sentence means. “The design was iteratively modified and rested over 8 weeks”. Please clarify and/or elaborate…”

- We have elaborated on and clarified this sentence. Page 11, lines 194-195.

“Page 10, line 198. It should be “The Fit-testing device” not the “The Fit-tester device”.”

- We have changed the object reference as requested. Page 11, line 200.

“Page 14, lines 271-274. Paragraph should be relocated to be the first paragraph after the subheading “QNFT performance during preliminary validation”.”

- By Editorial stipulations, PONE have asked that this paragraph be moved even higher in the Methods section. Apologies for the editorial stylistic difference. Page 11, lines 225-228.

“Page 14, lines 279-280. Although I appreciate that the authors included anthropometric data, they really ought to compare their test subjects with anthropometric data of typical respirator users as per Zhuang and Bradtmiller (2005) Head-and-face anthropometric survey of US respirator users. Journal of occupational and environmental hygiene 2(11):567-76. This comparison is important to allow the reader to ascertain if the novel respirator would be suitable for his/her workforce.”

- Yes, we agree. This is a weakness that has been addressed in our discussion-limitations, and a larger sample of healthcare workers will need to be tested. The follow-up paper (published in PONE and referenced in the revised MS) looks at a larger representative sample of face types. Page 21, lines 405-409.

“Figure 03. Text is not clear in the red boxes.”

- We have used white background to the text to make it clearer.

“S1 Table. What does “NIOSH Panel” refer to?”

- The NIOSH Panel has now been spelt out and referenced under S1 Table. Thanks for alerting us of this internally used shorthand reference.

---

Dear Reviewer 2:

- Thanks once again for the encouragement. This MS with minor revisions should have highlights in color this time in the tracked version.

We hope the minor revisions to the MS as indicated above has made this submission acceptable to PLOS, in line with the Journal’s academic standards. Thank you all once again for efforts in improving this MS.

Truly,

William C. K. Ng

MBBS MMed FANZCA FRCPC

Anesthesia and Pain Management, Toronto General Hospital

TGH Research Institute & UHN Advanced Perioperative Imaging Lab

Pediatric Cardiac Anesthesia, SickKids Hospital

Faculty Dept. of Anesthesiology, University of Toronto

M 1.857.330.7399

E William.Ng@uhn.ca or William.Ng@sickkids.ca

Soli Deo Gloria

---

## [Editor Report · Decision Letter 2]

10 Feb 2021

Development, manufacturing, and preliminary validation of a reusable half-face respirator during the COVID-19 pandemic

PONE-D-20-21977R2

Dear Dr. William C.K. Ng,

We’re pleased to inform you that your manuscript has been judged scientifically suitable for publication and will be formally accepted for publication once it meets all outstanding technical requirements.

Kind regards,

Tommaso Lomonaco, Ph.D

Academic Editor

PLOS ONE

Additional Editor Comments (optional):

Dear Authors, all the questions were answered and thus I suggest to accept the manuscript.

Regards,

Tommaso Lomonaco

---

## [Editor Report · Acceptance letter]

12 Mar 2021

PONE-D-20-21977R2 

Development, manufacturing, and preliminary validation of a reusable half-face respirator during the COVID-19 pandemic

Dear Dr. Ng:

I'm pleased to inform you that your manuscript has been deemed suitable for publication in PLOS ONE. Congratulations! Your manuscript is now with our production department. 

Kind regards, 

on behalf of

Dr. Tommaso Lomonaco 

Academic Editor

PLOS ONE